# Ultra-Wideband Narrow Wall Waveguide-to-Microstrip Transition Using Overlapped Patches

**DOI:** 10.3390/s22082964

**Published:** 2022-04-12

**Authors:** Ivan Zhou, Jordi Romeu Robert

**Affiliations:** School of Telecommunication Engineering, Universitat Politècnica de Catalunya, 08034 Barcelona, Spain; jordi.romeu-robert@upc.edu

**Keywords:** ultra-wideband, millimeter-wave, beamforming

## Abstract

An ultrawideband rectangular waveguide to microstrip line transition operating at the whole LMDS and Ka band is presented. The transition is based on exciting three overlapped transversal patches that radiate into the narrow wall of the waveguide, making the design feasible to be used in $\lambda_{g}/2$ spaced phased arrays. Both top-side and bottom-side versions were designed and compared to show their differences. They were validated by means of a manufactured back-to-back (B2B) configuration, with a measured fractional bandwidth of 21.2% (top-side) and 23% (bottom-side). The maximum single transition measured insertion losses were 0.67 dB (top-side) and 0.85 dB (bottom-side) in the whole band of operation.

## 1. Introduction

With the upcoming 5G millimeter-wave (mmWave) communication, a clear enhancement of the data speed, latency, and network efficiency is expected [1]. Transmission losses at these frequency bands are high [2,3], which requires the use of low loss transmission lines and antennas such as rectangular waveguide (RW) and horns, for example [4]. In addition, packaging and integration of low loss antennas into the RF chipsets or connectors often require microstrip-to-waveguide transitions, making the design more integrated and compact [5,6].

There are three main type of transitions: inline [7,8], bottom-side (BS) [9,10,11], and top-side (TS) [12,13,14,15,16]. For the case of inline transitions, the direction of propagation of the fields for both microstrip line (ML) and RW are the same, whereas for the TS and BS, their directions are perpendicular. In the case of TS, the ML and the RW are on the same face of the substrate, whereas for the BS the ML and the RW are on opposite faces, which may be necessary depending on the design requirements.

Inline transitions have the advantage of offering huge bandwidth with low insertion losses (IL). In [8], a tapered ridge along the axis of the RW was proposed, covering the whole Ka-band, but it requires the RW to be fabricated in two separate pieces and the substrate to be suspended in the air; in order to guarantee a good performance, a solid substrate must be implemented. In [10], a BS transition was proposed via proximity coupling through a patch antenna, although it offers 18% of fractional bandwidth (FBW), it requires perpendicular input/output ports from the large wall of the RW, which makes the design bulkier. In [11], this FBW is highly increased to 33.3% by means of an E-plane probe, but side feeding is needed, which also shows the same disadvantages as inline transitions. In [14], this large wall is avoided with a narrow-wall design consisting of a V-shaped aperture coupled patch; however, only 7.5% of FBW is achieved. TS RW to ML transitions are also explored as in [15], where the RW is excited through a transversal patch antenna coming from the narrow wall; however, intrusion elements inside the RW have to be carefully inserted in order to enhance the FBW from 11% to 15%, which increases the fabrication complexity. This intrusion element is avoided in [16] through a patch fed by a coupled ML, but only 11% of FBW is achieved.

In comparison with the transitions presented in this paper, a narrow-wall ML-RW transition is designed and validated using a manufactured back-to-back (B2B) configuration that can work for both TS and BS at the same time without increased manufacturing complexity. The transition consists of an ML that feeds an array of three overlapped transversal patches from the narrow wall of the RW, offering a much higher bandwidth when compared to similar presented work. The design shows a FBW of 21.2% for the TS design and 23% for the BS design.

## 2. Transition Design

The focus of this first section consists of the conversion of the Q-TEM mode of a 50 Ω ML [17] to the TE01 mode of a WR-34 ($W_{rw}\times L_{rw}$) [18]; a detailed description of the working principle will be given in Section 2.1 for the case of the TS transition where the RW is located on the same side of the substrate as the ML. In Section 2.2, a BS transition using the same working principle will be presented by just adding a metallic bias to connect the TS ML with the BS ML. Both TS and BS transitions will be compared in Section 3 to show the advantages and disadvantages of each type of transition, as depending on each design requirement, TS or BS will be needed.

### 2.1. Top-Side Transition

Figure 1a shows the isometric view of the transition, and Figure 1b shows its top and bottom view. The transition is designed using a 0.81 mm thick RO4003 substrate with a dielectric constant of Er = 3.55 and a loss tangent of 0.002.

A full-wave electromagnetic simulation tool, CST Microwave Studio, is used for its optimization. In order to excite the TE01 mode from the waveguide, excitation of transversal currents in the $\hat{x}$ direction is needed. A simple and direct ML-RW mode conversion can be found in [15]. By using a single transversal patch with an approximated length of $\frac{1}{2}\lambda_{g}$, radiation with the proper polarization is achieved to excite the TE01 mode from the RW. However, it is bandwidth limited. As a way to enhance this bandwidth, an array of three overlapped transversal patches is proposed in this paper, the length Lp of the patches, the width $W_{p}$, inter-element distance $d_{y}$ and array position with respect to the entrance of the RW $y_{0}$ are jointly optimized to provide maximum bandwidth. The overlapping of these $0.2\lambda_{g}$ length patches is crucial for the excitation of transversal $J\hat{x}$ currents. In Figure 2 we show a sketch of the current distribution at the central frequency of 26.5 GHz along the patches (a) without the middle patch (no overlapping) and (b) with the middle patch (with overlapping). As we can see in Figure 2a, which is the transition with the parameters from Table 1 but without overlapping, the $J\hat{x}$ transversal currents are not excited by the patches at the central frequency $f_{0}$, because the effective length of the patches is not close to $\frac{1}{2}\lambda_{g}$.

With the introduction of a central patch element that overlaps with the other two from Figure 2a (see Figure 2b), the path of the currents along the transversal $\hat{x}$ direction is effectively enlarged, from $0.2\lambda_{g}$ to $0.4\lambda_{g}$, exciting in phase $J\hat{x}$ currents at lower frequencies, so the Q-TEM mode from the ML is directly converted to the TE01 mode for the RW as a result of the radiation of the array into the RW. Note that the currents are diagonal, so the path that the currents are taking to travel is slightly larger than $0.4\lambda_{g}$. In Figure 3 we can see the resonance frequencies for both cases with the presence of the RW.

The overlapped patches offer an intrinsic higher bandwidth behaviour when compared to a single patch element. The array is fed by a ML of width $W_{f}$, and it is matched to a 50 Ω circuit with a stepped section of $W_{t}\times L_{t}$. The entrance to the RW has dimensions (Win×Hin), it should ideally be as little as possible to avoid leakage from the RW but not short-circuiting with the ML; however, a considerable dimension was chosen to be manufacturable. There are via holes of radius $r_{b}$ and periodicity $d_{b}$ surrounding the aperture of the RW that connects the RW to the ground plane of the substrate; by reducing this periodicity, the insertion losses are not reduced.

### 2.2. Bottom-Side Transition

The working principle of the BS transition is the same as the TS transition. The parameters are not exactly the same, as they were slightly re-optimized for this BS case, see Table 2. Figure 4 shows the top view, bottom view, and isometric view of this transition. The entrance of the ML along the bottom side is achieved by leaving a U-slot of width *c* around the ground. By adding a bias of radius $r_{b}$ at the edge of this ML, we can connect and feed the overlapped patches from the other side, the TS. The ground plane has an extension of $E_{x}$ with respect to the narrow wall RW. The via holes also have to be extended to the end of this ground plane, otherwise leakage through $\hat{x}$ direction will appear due to the currents excited by the U-slot.

This type of BS transition has an advantage with respect to the TS transitions involving interference reduction, as the RF-chipset may be located at the contrary face from the radiating elements. However, it comes with many disadvantages, as will be seen in the next section.

## 3. Simulated Results

Simulations involving a parametric study were carried out for both TS transition (see Figure 5) and BS transition (see Figure 6) to show a better understanding of the sensitivity of each parameter.

We can see that the input reflection coefficient for both designs are very similar, ranging approximately from 23.75 GHz to 30 GHz. However, the maximum insertion losses (IL) for the BS are slightly worse (−0.65 dB) than for the TS (−0.4 dB) along the frequency band of operation. For the case of TS transition, we can clearly see two main resonant frequencies at 24.8 GHz and 29 GHz. The lower resonance frequency is controlled by the patch’s length $L_{p}$, whereas the upper resonance frequency is controlled by the patch’s width $W_{p}$. Both resonant frequencies can be simultaneously shifted by adjusting the $d_{y}$. For the case of BS transition, $L_{p}$ also controls the lower resonance frequency, $W_{p}$ controls both resonances as well as $d_{y}$. We can also see that the sensitivity of the insertion losses against these parameter changes are much higher for the BS case; we see a maximum penalty of up to 0.4 dB for the $d_{y}-0.1$ mm trace, and only 0.2 dB for the $L_{p}+0.03$ mm trace from the TS case.

In Figure 7 we plot the leaked radiation for both TS and BS transitions. This simulation consists of placing both ML and RW ports and plotting the far-field when exciting the ML port. The leakage for the TS is −10.28 dB, whereas that for the BS is higher, peaking at −7.6 dB. In order to reduce this, the slot’s width *c* from this ground should be as small as possible; however, it would not be viable to be manufactured. Another way to reduce this leakage is by inserting a quarter wavelength metallic box to enclose this slot [19].

The TS transition shows better input reflection coefficients, insertion losses, and radiation leakage performance.

## 4. Assembly and Measurements

A B2B transition was designed and manufactured at the UPC facilities for both TS and BS designs. The PCB was fabricated using the standard photolithography technique, and the RW assembly connection was manufactured using a CNC milling machine (see Figure 8 and Figure 9). The two designs were measured with the vector network analyzer ZNB-40 from Rohde-Schwarz.

In Figure 10 and Figure 11, the 73 mm long B2B input reflection coefficient and IL from both simulations and measurements are shown for TS and BS designs, respectively. There is a good correlation between the simulations and measurements; for the case of TS transition the design offers a measured −10 dB bandwidth ranging from 24.25 GHz to 30 GHz (FBW of 21.2%) with a little portion of mismatched band ranging from 26.35 GHz to 26.65 GHz at just −9.5 dB and a maximum B2B IL of 1.35 dB, which is 0.67 dB (half) for a single transition. Note that the IL does not take into account the propagation losses (1.3 dB for the 73 mm long ML), as it was already taken away from both simulation and measurement plots. This portion of slightly mismatched frequencies is due to a positional error of the holes that are used to attach the WR-34 to the transition; obviously, the results would be much better if, instead of a B2B transition, a direct single ML-RW transition was used to be measured.

For the case of BS transition, the design offers a measured −10 dB bandwidth ranging from 24.6 GHz to 31 GHz (FBW of 23%) and a maximum IL of 1.7 dB, which is 0.85 dB for a single transition. This IL is worse than the simulated one because of slight manufacturing errors, as seen in the previous section, BS is less robust than TS.

Clearly, the TS offers much better and stable IL along the frequency band of operation. Although for both cases there is a small portion of slightly mismatched frequencies around the central frequency point, the TS design is a preferred option in terms of IL, simpler manufacturing process, and robustness to fabrication errors.

## 5. Comparison with Other Work

Table 3 shows the comparison with other related work. There are mainly three types of transitions as described in the introduction: inline, BS and TS. The wall column refers to the feeding of the ML through the RW, whereas the back-short column refers to whether a back-shorting cavity is needed. Inline transitions [8] provide much higher FBW (33.3%) with low IL (0.6 dB) with the cost of having to manufacture the RW in two separate pieces as the ML is inserted along the axis of the RW, so it remains floating inside the RW, which requires solid substrates. Although it has brilliant performance, the direction of propagation of the fields along the RW is aligned with the one in the ML, which may not be useful depending on the design requirement.

Perpendicularity between the direction of propagation of the fields in the ML and the RW are often required, making the designs easier to manufacture, for example, horn antennas that are usually placed above the PCB. That is why TS and BS transitions may be preferred. In [10], a broad wall BS transition was designed with an FBW of 18%, but it is bulky in the transversal direction of the ML, which may be inconvenient for many applications related to phased arrays, for example. TS or BS narrow wall transitions are mainly limited by the bandwidth. The highest one found in the literature reaches only up to 15% of FBW for the TS and only 7.5% for the BS. In both designs of our work, we achieved a much higher FBW, 21.2% and 23%, respectively. Although for the BS case, the IL is just 0.15 dB worse, the TS case is greater for both IL and FBW.

## 6. Conclusions

A novel waveguide to microstrip transition using an array of three overlapped patches has been designed for its integration with the RF chipset. Two designs were designed and manufactured using the same principle for both TS and BS integration. The TS B2B transition offers a measured 21.2% of FBW with a maximum single transition IL of up to 0.67 dB. The BS B2B transition offers 23% of FBW with a maximum single transition IL of up to 0.85 dB. Both TS and BS transitions offers a similar input reflection coefficient, but a higher radiation leakage is found for the BS (2.7 dB more).

The transition is suitable to be used for the whole LMDS and K-band for 5G millimeter-wave applications requiring low cost and high bandwidth for the integration of circuits/antennas requiring RW interfaces, or RW-antennas requiring feeding from ML, such as horn antennas.

## Figures and Tables

**Figure 1 sensors-22-02964-f001:**
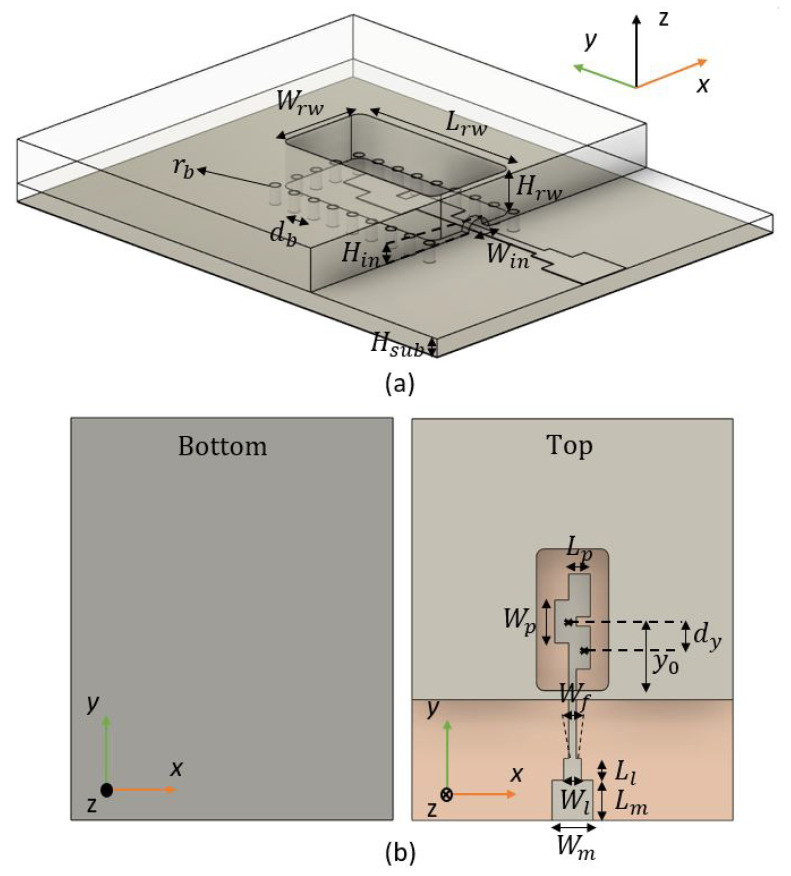
Artist view of the Top-Side transition for (**a**) isometric view and (**b**) top and bottom view.

**Figure 2 sensors-22-02964-f002:**
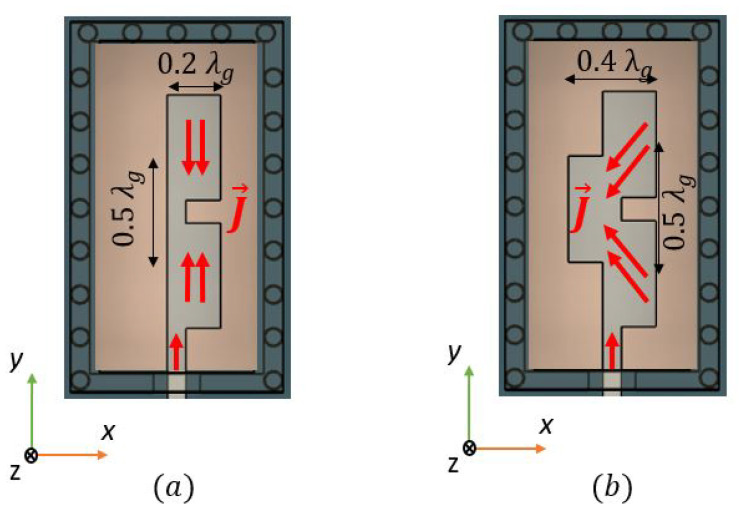
Top view of the current distribution at $f_{0}$ = 26.5 GHz of (**a**) the transition without overlapping, (**b**) the transition with overlapping.

**Figure 3 sensors-22-02964-f003:**
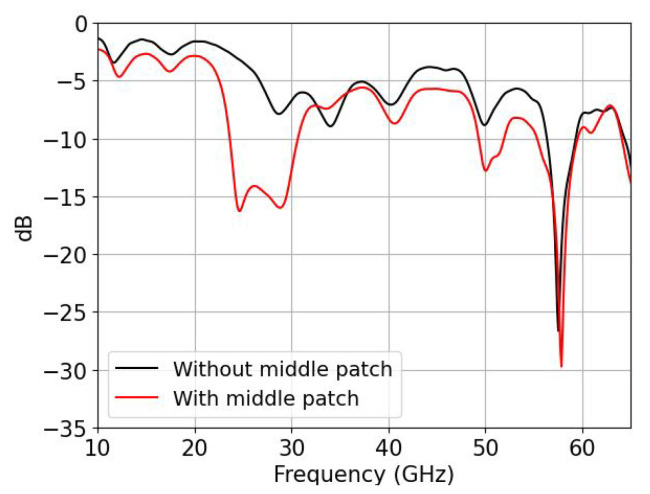
Input reflection coefficients without overlapping (black) and with overlapping (red).

**Figure 4 sensors-22-02964-f004:**
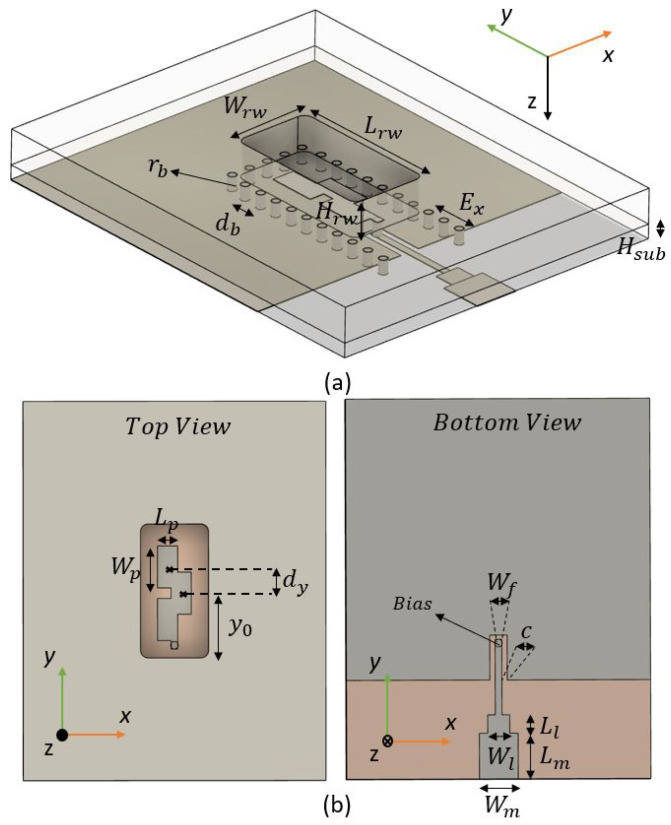
Artist view of the Bottom-Side transition for (**a**) isometric view and (**b**) top and bottom view.

**Figure 5 sensors-22-02964-f005:**
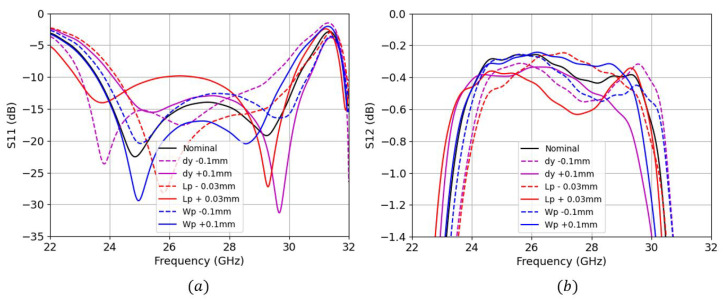
Top-side parametric study of S-parameters (**a**) input reflection coefficient (**b**) insertion losses.

**Figure 6 sensors-22-02964-f006:**
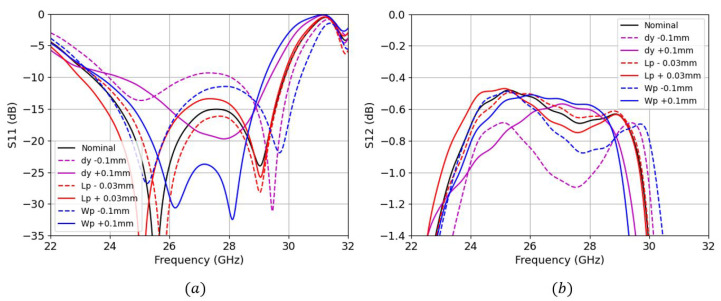
Bottom-side parametric study of S-parameters (**a**) input reflection coefficient (**b**) insertion losses.

**Figure 7 sensors-22-02964-f007:**
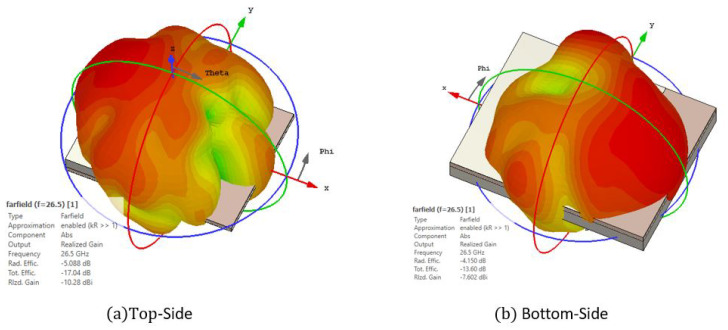
Simulated radiation leakage for the nominal case for both (**a**) TS and (**b**) BS transitions.

**Figure 8 sensors-22-02964-f008:**
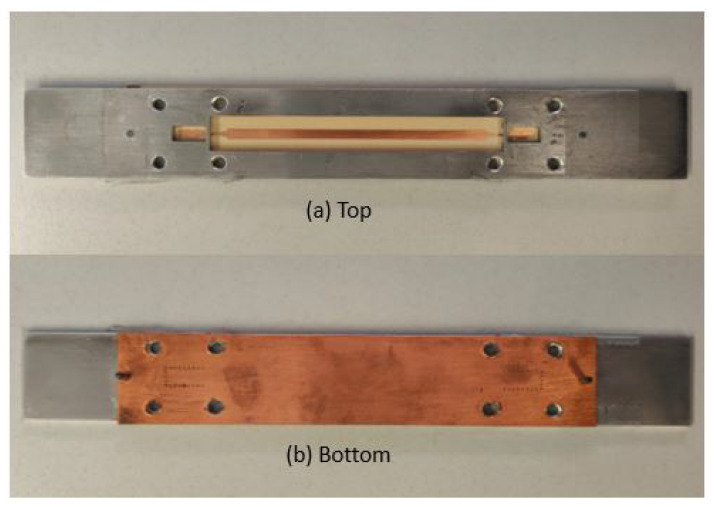
Top view and bottom view of the B2B manufactured TS transition.

**Figure 9 sensors-22-02964-f009:**
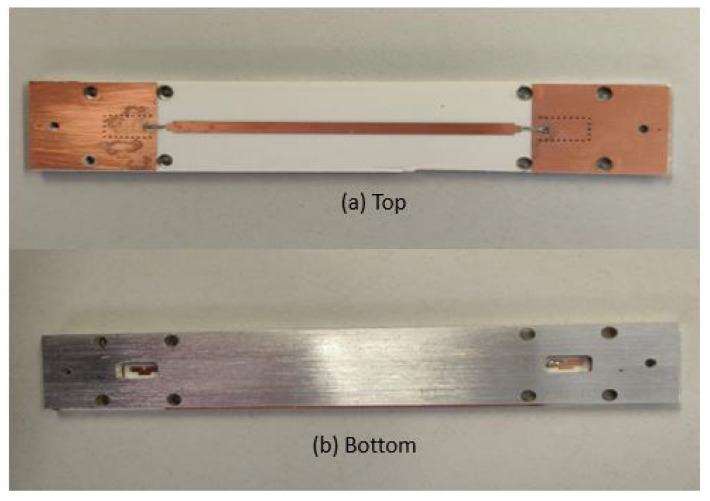
Top view and bottom view of the B2B manufactured BS transition.

**Figure 10 sensors-22-02964-f010:**
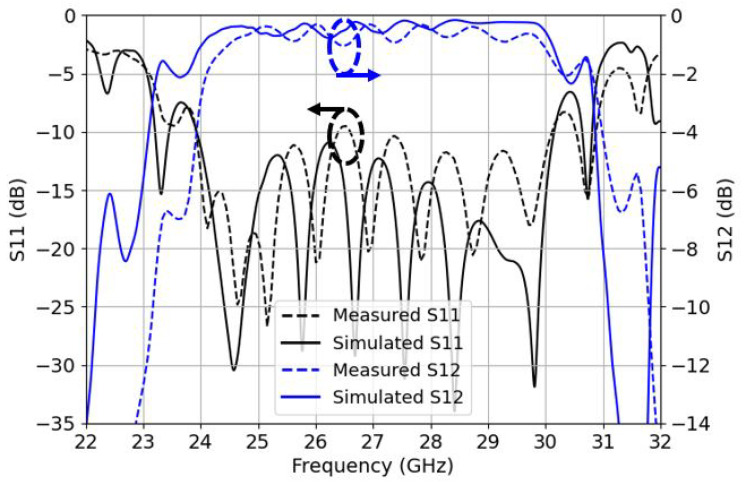
Top-side back-to-back S11 reflection coefficients (left *y*-axis) in black and insertion losses S12 (right *y*-axis) in blue for the measurements and simulations.

**Figure 11 sensors-22-02964-f011:**
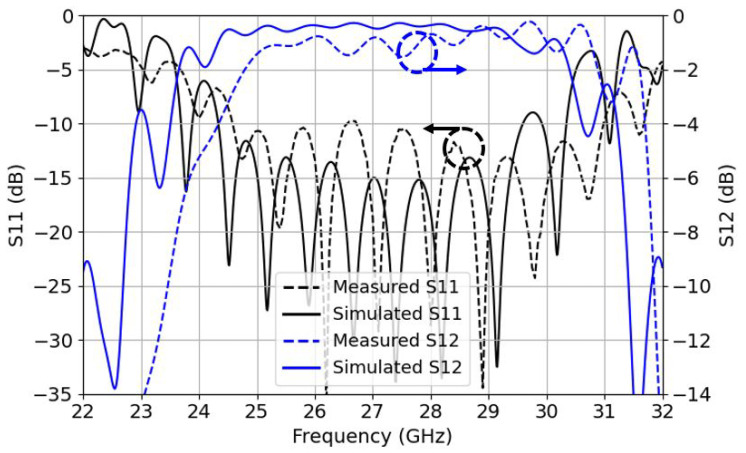
Bottom-side back-to-back S11 reflection coefficients (left *y*-axis) in black and insertion losses S12 (right *y*-axis) in blue for the measurements and simulations.

**Table 1 sensors-22-02964-t001:** Dimensions in millimeters of each designed parameter for the top-side transition.

$H_{sub}$	0.81	$W_{f}$	0.47	$W_{rw}$	4.5
dy	1.625	$W_{l}$	1.1	$L_{rw}$	8.8
$W_{p}$	2.67	$L_{l}$	1.35	$H_{rw}$	2
$L_{p}$	1.35	$L_{m}$	2.5	$H_{in}$	0.5
$Y_{0}$	4.3	$W_{m}$	2.55	$W_{in}$	1.2
$r_{b}$	0.25	$d_{b}$	1.17		

**Table 2 sensors-22-02964-t002:** Dimensions in millimeters of each designed parameter for the bottom-side transition.

$H_{sub}$	0.81	$W_{f}$	0.44	$W_{m}$	2.55	$Y_{0}$	4.4
dy	1.73	$W_{l}$	1.45	$L_{m}$	2.5	$r_{b}$	0.25
$W_{p}$	2.77	$L_{l}$	1.2	$H_{rw}$	2	$W_{rw}$	4.5
$L_{p}$	1.33	$d_{b}$	1.17	*C*	0.15	$L_{rw}$	8.8
$E_{x}$	1.58						

**Table 3 sensors-22-02964-t003:** Similar work that can be found showing the comparison of B2B parameters like minimum IL, maximum IL, FBW, and the operating band.

Work	Type	Wall	Max IL	FBW	Band	Backshort
[8]	Inline	Aligned	0.6 dB	33.3%	Ka	Yes
[10]	BS	Broad	NA	18%	W	No
[14]	BS	Narrow	0.7 dB	7.5%	W	No
This Work BS	BS	Narrow	0.85 dB	23%	LMDS, Ka	No
[15] (A)	TS	Narrow	1 dB	15%	W	No
[15] (B)	TS	Narrow	1.1 dB	11%	W	No
[16]	TS	Narrow	NA	11%	W	No
This Work TS	TS	Narrow	0.67 dB	21.2%	LMDS, Ka	No

## Data Availability

Not applicable.

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
