# Peer review of "Ultra-Wideband Narrow Wall Waveguide-to-Microstrip Transition Using Overlapped Patches"

_sensors, 2022, doi:10.3390/s22082964_

Round 1

Reviewer 1 Report

In this paper a WG-to-microstrip transition is presented.Some major issues have been identified:

1) In most applications where such a component is required hermeticity is a key requirement. However this aspect is completely ignored in this work.

2) discrepancy between simulation and measurement are quite significant ans should be addressed in much deeper detail. In particular an analysis on the transition efficiency should be performed, with the aim to identify the source of additional losses, which cannot be arbitrarily attributed to misalignments in the manufacturing. A strong leakage issue is most probably present and not addressed, which could justify the considerable increase of IL wrt simulation, possibly associated to some internal resonances at specific frequencies. |S11|^2 + |S21|^2 would give an estimate of total loss in the circuit - a subsequent removal of microstrip loss would provide total transition efficiency.

3) IL performance is already quite poor from simulated results, with ripples in the operative band which would be unacceptable in many applications - design must be definitely improved by minimizing IL ripples and improving RL.

Author Response

Dear reviewer, please find the response to your comments in pdf format. The revised article is also included in the same pdf. 

Kind regards,

Ivan

Reviewer 2 Report

A microstripline-to-waveguide transition is presented in this paper. The comments are as following:

  1. What is the difference between the top-side (TS) and bottom-side (BS) versions of your transition design? Why we need to compare this two versions?
  2. Why choose such a shape of the patch in the transition part? Does it has any advantage compared with a regular-shaped patch?
  3. As in Fig. 11, the measured insertion loss of the BS version is higher than 5 dB. Why is it stated that “…… maximum IL of 2.7dB.” in the paper?
  4. What is the application of such proposed design?
  5. The references in the Tab. 3 are from 10 years ago, can you provide any state-of-the-art papers in the recent 5 years?

Author Response

Dear reviewer, please find the responses to your comments in pdf format. The revised article is also included in the same pdf.

Kind regards,

Ivan

Reviewer 3 Report

The paper is interesting and the implementation is very detailed. However, the authors are comparing their implementation with some relevant designs that are almost 10 years old (the most recent is in 2013). Consequently, some more recent ones must be included such as (but not limited to):
1) Millimeter-Wave Waveguide-to-Microstrip Transition With a Built-In DC/IF Return Path
2) Full Ka Band Waveguide-to-Microstrip Inline Transition Design

Some more comments:

  1. The acronyms should be described in the main text, not in the abstract (e.g. RW).

  2. The contribution of this work in the introduction is very brief. In particular, it is a single sentence (lines 29-32) that is not easily comprehensible.

  3. The manipulation of the English language is very bad.

  4. The resonance frequencies in Fig. 3 correspond to free-space radiation (namely the waveguide is absent)?

  5. Figures 10 and 11 have no reference in their caption concerning the device.

  6. The results of the proposed arrangements and their comparison to literature are very limited. The discussion must be extended.

Author Response

(The authors gave the same response as above.)

Reviewer 4 Report

This work discuss about the transition from transmission line to waveguide for both top side and bottom side configurations. The designs are both simulated and fabricated which was very interesting. Good job authors! However I had some comments/suggestions which can make the manuscript richer.   

1- Line7: When you provided references for TS and BS transition, you only mentioned 1 or 2 references for each. However in the rest of the manuscript you brought up many other references. You better mention them in the introduction line 7. 

2- Figure1: The via spacing seems close to quarter wavelength, which should be good enough for not letting the wave/energy scattered to the side. Did you try shorter spacing (like lambda/8) too to see the differences. 

3- P2, Line56: When you mentioned "... optimized transition without overlapping...", what you specifically meant by "optimized". Did you run any optimization algorithm? 

4- P2, Line56-58: The last sentence of the paragraph is not well understandable to me. 

5- P2, Line 61. You mentioned the effective width along x-axis is 0.4λg. Couldn't you make it 0.5 wavelength? What would be the downside of half wavelength effect width?

6- Line 67: when you mentioned "Wt xLt " you probably meant Wl xLl since we see Wl and Lin the figures. 

7- Figure 6. It will be more helpful if you put markers on the plots so they are distinguishable. 

8- Page 5. Line98 and after: I dont understand the way you check the leakage. Mostly when you are going to check the far field you need to put something like PML at the second port, and then excite the ML and see how much the power with the radiated (total radiated power). It seems you didn't do that and the second port (which is related waveguide) is not there. 

9- Page 9. Ref 11 is not well typed. 

Thanks again. 

Author Response

Dear reviewer, please find the responses to your comments in pdf format (reviewer_4.pdf). The revised article is also included in the same pdf.

Kind regards,

Ivan

Round 2

Reviewer 3 Report

The authors conducted the majority of the proposed changes. Although some aspects, especially in the introduction, can be still improved, the paper is in a good shape for publication.